# VARIATIONAL RECURRENT MODELS FOR SOLVING PARTIALLY OBSERVABLE CONTROL TASKS

**Dongqi Han**
Cognitive Neurorobotics Research Unit
Okinawa Institute of Science and Technology
Okinawa, Japan
dongqi.han@oist.jp

**Kenji Doya**
Neural Computation Unit
Okinawa Institute of Science and Technology
Okinawa, Japan
doya@oist.jp

**Jun Tani**[*]
Cognitive Neurorobotics Research Unit
Okinawa Institute of Science and Technology
Okinawa, Japan
jun.tani@oist.jp

## ABSTRACT

In partially observable (PO) environments, deep reinforcement learning (RL) agents often suffer from unsatisfactory performance, since two problems need to be tackled together: how to extract information from the raw observations to solve the task, and how to improve the policy. In this study, we propose an RL algorithm for solving PO tasks. Our method comprises two parts: a variational recurrent model (VRM) for modeling the environment, and an RL controller that has access to both the environment and the VRM. The proposed algorithm was tested in two types of PO robotic control tasks, those in which either coordinates or velocities were not observable and those that require long-term memorization. Our experiments show that the proposed algorithm achieved better data efficiency and/or learned more optimal policy than other alternative approaches in tasks in which unobserved states cannot be inferred from raw observations in a simple manner[1].

## 1 INTRODUCTION

Model-free deep reinforcement learning (RL) algorithms have been developed to solve difficult control and decision-making tasks by self-exploration (Sutton & Barto, 1998; Mnih et al., 2015; Silver et al., 2016). While various kinds of fully observable environments have been well investigated, recently, partially observable (PO) environments (Hafner et al., 2018; Igl et al., 2018; Lee et al., 2019; Jaderberg et al., 2019) have commanded greater attention, since real-world applications often need to tackle incomplete information and a non-trivial solution is highly desirable.

There are many types of PO tasks; however, those that can be solved by taking the history of observations into account are more common. These tasks are often encountered in real life, such as videos games that require memorization of previous events (Kapturowski et al., 2018; Jaderberg et al., 2019) and robotic control using real-time images as input (Hafner et al., 2018; Lee et al., 2019). While humans are good at solving these tasks by extracting crucial information from the past observations, deep RL agents often have difficulty acquiring satisfactory policy and achieving good data efficiency, compared to those in fully observable tasks (Hafner et al., 2018; Lee et al., 2019).

For solving such PO tasks, several categories of methods have been proposed. One simple, straightforward solution is to include a history of raw observations in the current "observation" (McCallum, 1993; Lee et al., 2019). Unfortunately, this method can be impractical when decision-making requires a long-term memory because dimension of observation become unacceptably large if a long history is included.

---

[*]Corresponding author.
[1]Codes are available at https://github.com/oist-cnru/Variational-Recurrent-Models.

Another category is based on model-free RL methods with recurrent neural networks (RNN) as function approximators (Schmidhuber, 1990; 1991; Igl et al., 2018; Kapturowski et al., 2018; Jaderberg et al., 2019), which is usually more tractable to implement. In this case, RNNs need to tackle two problems simultaneously (Lee et al., 2019): learning representation (encoded by hidden states of the RNN) of the underlying states of the environment from the state-transition data, and learning to maximize returns using the learned representation. As most RL algorithms use a bootstrapping strategy to learn the expected return and to improve the policy (Sutton & Barto, 1998), it is challenging to train the RNN stably and efficiently, since RNNs are relatively more difficult to train (Pascanu et al., 2013) than feedforward neural networks.

The third category considers learning a model of the environment and estimating a *belief state*, extracted from a sequence of state-transitions (Kaelbling et al., 1998; Ha & Schmidhuber, 2018; Lee et al., 2019). The belief state is an agent-estimated variable encoding underlying states of the environment that determines state-transitions and rewards. Perfectly-estimated belief states can thus be taken as "observations" of an RL agent that contains complete information for solving the task. Therefore, solving a PO task is segregated into a representation learning problem and a fully observable RL problem. Since fully observable RL problems have been well explored by the RL community, the critical challenge here is how to estimate the belief state.

In this study, we developed a variational recurrent model (VRM) that models sequential observations and rewards using a latent stochastic variable. The VRM is an extension of the variational recurrent neural network (VRNN) model (Chung et al., 2015) that takes actions into account. Our approach falls into the third category by taking the internal states of the VRM together with raw observations as the belief state. We then propose an algorithm to solve PO tasks by training the VRM and a feed-forward RL controller network, respectively. The algorithm can be applied in an end-to-end manner, without fine tuning of a hyperparameters.

We then experimentally evaluated the proposed algorithm in various PO versions of robotic control tasks. The agents showed substantial policy improvement in all tasks, and in some tasks the algorithm performed essentially as in fully observable cases. In particular, our algorithm demonstrates greater performance compared to alternative approaches in environments where only velocity information is observable or in which long-term memorization is needed.

## 2 RELATED WORK

Typical model-based RL approaches utilize learned models for dreaming, i.e. generating state-transition data for training the agent (Deisenroth & Rasmussen, 2011; Ha & Schmidhuber, 2018; Kaiser et al., 2019) or for planning of future state-transitions (Watter et al., 2015; Hafner et al., 2018; Ke et al., 2019). This usually requires a well-designed and finely tuned model so that its predictions are accurate and robust. In our case, we do not use VRMs for dreaming and planning, but for auto-encoding state-transitions. Actually, PO tasks can be solved without requiring VRMs to predict accurately (see Appendix E). This distinguishes our algorithm from typical model-based RL methods.

The work our method most closely resembles is known as *stochastic latent actor-critic* (SLAC, Lee et al. (2019)), in which a latent variable model is trained and uses the latent state as the belief state for the critic. SLAC showed promising results using pixels-based robotic control tasks, in which velocity information needs to be inferred from third-person images of the robot. Here we consider more general PO environments in which the reward may depend on a long history of inputs, e.g., in a snooker game one has to remember which ball was potted previously. The actor network of SLAC did not take advantage of the latent variable, but instead used some steps of raw observations as input, which creates problems in achieving long-term memorization of reward-related state-transitions. Furthermore, SLAC did not include raw observations in the input of the critic, which may complicate training the critic before the model converges.

## 3 BACKGROUND

### 3.1 PARTIALLY OBSERVABLE MARKOV DECISION PROCESSES

The scope of problems we study can be formulated into a framework known as *partially observable Markov decision processes* (POMDP) (Kaelbling et al., 1998). POMDPs are used to describe decision or control problems in which a part of underlying states of the environment, which determine state-transitions and rewards, cannot be directly observed by an agent.

A POMDP is usually defined as a 7-tuple $(\mathbb{S}, \mathbb{A}, T, R, \mathbb{X}, O, \gamma)$, in which $\mathbb{S}$ is a set of states, $\mathbb{A}$ is a set of actions, and $T : \mathbb{S} \times \mathbb{A} \to p(\mathbb{S})$ is the state-transition probability function that determines the distribution of the next state given current state and action. The reward function $R : \mathbb{S} \times \mathbb{A} \to \mathbb{R}$ decides the reward during a state-transition, which can also be probabilistic. Moreover, $\mathbb{X}$ is a set of observations, and observations are determined by the observation probability function $O : \mathbb{S} \times \mathbb{A} \to p(\mathbb{X})$. By defining a POMDP, the goal is to maximize expected discounted future rewards $\sum_t \gamma^t r_t$ by learning a good strategy to select actions (policy function).

Our algorithm was designed for general POMDP problems by learning the representation of underlying states $s_t \in \mathbb{S}$ via modeling observation-transitions and reward functions. However, it is expected to work in PO tasks in which $s_t$ or $p(s_t)$ can be (at least partially) estimated from the history of observations $x_{1:t}$.

### 3.2 VARIATIONAL RECURRENT NEURAL NETWORKS

To model general state-transitions that can be stochastic and complicated, we employ a modified version of the VRNN (Chung et al., 2015). The VRNN was developed as a recurrent version of the variational auto-encoder (VAE, Kingma & Welling (2013)), composed of a variational generation model and a variational inference model. It is a recurrent latent variable model that can learn to encode and predict complicated sequential observations $x_t$ with a stochastic latent variable $z_t$.

The generation model predicts future observations given the its internal states,

$$z_t \sim \mathcal{N}\left(\boldsymbol{\mu}_{p,t}, \mathrm{diag}(\boldsymbol{\sigma}_{p,t}^2)\right), \quad \left[\boldsymbol{\mu}_{p,t}, \boldsymbol{\sigma}_{p,t}^2\right] = f^{\mathrm{prior}}(\boldsymbol{d}_{t-1}),$$
$$x_t | z_t \sim \mathcal{N}\left(\boldsymbol{\mu}_{y,t}, \mathrm{diag}(\boldsymbol{\sigma}_{y,t}^2)\right), \quad \left[\boldsymbol{\mu}_{y,t}, \boldsymbol{\sigma}_{y,t}^2\right] = f^{\mathrm{decoder}}(\boldsymbol{z}_t, \boldsymbol{d}_{t-1}), \tag{1}$$

where $f$s are parameterized mappings, such as feed-forward neural networks, and $\boldsymbol{d}_t$ is the state variable of the RNN, which is recurrently updated by

$$\boldsymbol{d}_t = f^{\mathrm{RNN}}(\boldsymbol{d}_{t-1}; \boldsymbol{z}_t, \boldsymbol{x}_t). \tag{2}$$

The inference model approximates the latent variable $z_t$ given $x_t$ and $\boldsymbol{d_t}$.

$$z_t | x_t \sim \mathcal{N}\left(\boldsymbol{\mu}_{z,t}, \mathrm{diag}(\boldsymbol{\sigma}_{z,t}^2)\right), \text{ where } \left[\boldsymbol{\mu}_{z,t}, \boldsymbol{\sigma}_{z,t}^2\right] = f^{\mathrm{encoder}}(\boldsymbol{x}_t, \boldsymbol{d}_{t-1}). \tag{3}$$

For sequential data that contain $T$ time steps, learning is conducted by maximizing the evidence lower bound $ELBO$, like that in a VEA (Kingma & Welling, 2013), where

$$ELBO = \sum_t^T \left[-D_{KL}(q(\boldsymbol{z}_t | \boldsymbol{z}_{1:t-1}, \boldsymbol{x}_{1:t}) || p(\boldsymbol{z}_t | \boldsymbol{z}_{1:t-1}, \boldsymbol{x}_{1:t-1}))\right]$$
$$+ \mathbb{E}_{q(\boldsymbol{z}_t | \boldsymbol{x}_{1:t}, \boldsymbol{z}_{1:t-1})}\left[\log\left(p(\boldsymbol{x}_t | \boldsymbol{z}_{1:t}, \boldsymbol{x}_{1:t-1})\right)\right], \tag{4}$$

where $p$ and $q$ are parameterized PDFs of $z_t$ by the generative model and the inference model, respectively. In a POMDP, a VRNN can be used to model the environment and to represent underlying states in its state variable $\boldsymbol{d}_t$. Thus an RL agent can benefit from a well-learned VRNN model since $\boldsymbol{d}_t$ provides additional information about the environment beyond the current raw observation $x_t$.

### 3.3 SOFT ACTOR CRITIC

*Soft actor-critic* (SAC) is a state-of-the-art model-free RL that uses experience replay for dynamic programming, which been tested on various robotic control tasks and that shows promising performance (Haarnoja et al., 2018a;b). A SAC agent learns to maximize reinforcement returns as well as entropy of its policy, so as to obtain more rewards while keeping actions sufficiently stochastic.

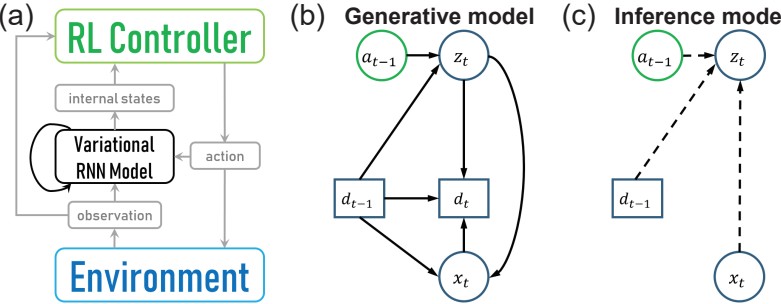

Figure 1: Diagrams of the proposed algorithm. **(a)** Overview. **(b, c)** The generative model and the inference model of a VRM.

A typical SAC implementation can be described as follows. The state value function $V(\boldsymbol{s})$, the state-action value function $Q(\boldsymbol{s}, \boldsymbol{a})$ and the policy function $\pi(\boldsymbol{a}|\boldsymbol{s})$ are parameterized by neural networks, indicated by $\psi, \lambda, \eta$, respectively. Also, an entropy coefficient factor (also known as the temperature parameter), denoted by $\alpha$, is learned to control the degree of stochasticity of the policy. The parameters are learned by simultaneously minimizing the following loss functions.

$$J_V(\psi) = \mathbb{E}_{\boldsymbol{s}_t \sim \mathcal{B}} \left[ \frac{1}{2} \left( V_\psi(\boldsymbol{s}_t) - \mathbb{E}_{\boldsymbol{a}_t \sim \boldsymbol{\pi}_\eta} \left[ Q_\lambda(\boldsymbol{s}_t, \boldsymbol{a}_t) - \alpha \log \pi_\eta(\boldsymbol{a}_t|\boldsymbol{s}_t) \right] \right)^2 \right], \quad (5)$$

$$J_Q(\lambda) = \mathbb{E}_{(\boldsymbol{s}_t, \boldsymbol{a}_t) \sim \mathcal{B}} \left[ \frac{1}{2} \left( Q_\lambda(\boldsymbol{s}_t, \boldsymbol{a}_t) - \left( r(\boldsymbol{s}_t, \boldsymbol{a}_t) + \gamma \mathbb{E}_{\boldsymbol{s}_{t+1} \sim \mathcal{B}} [V_\psi(\boldsymbol{s}_{t+1})] \right) \right)^2 \right], \quad (6)$$

$$J_\pi(\eta) = \mathbb{E}_{\boldsymbol{s}_t \sim \mathcal{B}} \left[ \mathbb{E}_{\boldsymbol{a}_\eta(\boldsymbol{s}_t) \sim \pi_\eta(\boldsymbol{s}_t)} \left[ \alpha \log \pi_\eta \left( \boldsymbol{a}_\eta(\boldsymbol{s}_t)|\boldsymbol{s}_t \right) - Q_\lambda(\boldsymbol{s}_t, \boldsymbol{a}_\eta(\boldsymbol{s}_t)) \right] \right], \quad (7)$$

$$J(\alpha) = \mathbb{E}_{\boldsymbol{s}_t \sim \mathcal{B}} \left[ \mathbb{E}_{\boldsymbol{a} \sim \pi_\eta(\boldsymbol{s}_t)} \left[ -\alpha \log \pi_\eta(\boldsymbol{a}|\boldsymbol{s}_t) - \alpha \mathcal{H}_{\text{tar}} \right] \right], \quad (8)$$

where $\mathcal{B}$ is the replay buffer from which $s_t$ is sampled, and $\mathcal{H}_{\text{tar}}$ is the target entropy. To compute the gradient of $J_\pi(\eta)$ (Equation. 7), the reparameterization trick (Kingma & Welling, 2013) is used on action, indicated by $\boldsymbol{a}_\eta(\boldsymbol{s}_t)$. Reparameterization of action is not required in minimizing $J(\alpha)$ (Equation. 8) since $\log \pi_\eta(\boldsymbol{a}|\boldsymbol{s}_t)$ does not depends on $\alpha$.

SAC was originally developed for fully observable environments; thus, the raw observation at the current step $\boldsymbol{x}_t$ was used as network input. In this work, we apply SAC in PO tasks by including the state variable $\boldsymbol{d}_t$ of the VRNN in the input of function approximators of both the actor and the critic.

## 4 METHODS

### 4.1 VARIATIONAL RECURRENT STATE-TRANSITION MODELS

An overall diagram of the proposed algorithm is summarized in Fig. 1(a), while a more detailed computational graph is plotted in Fig. 2. We extend the original VRNN model (Chung et al., 2015) to the proposed VRM model by adding action feedback, i.e., actions taken by the agent are used in the inference model and the generative model. Also, since we are modeling state-transition and reward functions, we include the reward $r_{t-1}$ in the current raw observation $\boldsymbol{x}_t$ for convenience. Thus, we have the inference model (Fig. 1(c)), denoted by $\phi$, as

$$\boldsymbol{z}_{\phi,t}|\boldsymbol{x}_t \sim \mathcal{N} \left( \boldsymbol{\mu}_{\phi,t}, \text{diag}(\boldsymbol{\sigma}_{\phi,t}^2) \right), \text{ where } \left[ \boldsymbol{\mu}_{\phi,t}, \boldsymbol{\sigma}_{\phi,t}^2 \right] = \phi(\boldsymbol{x}_t, \boldsymbol{d}_{t-1}, \boldsymbol{a}_{t-1}), \quad (9)$$

The generative model (Fig. 1(b)), denoted by $\theta$ here, is

$$\boldsymbol{z}_t \sim \mathcal{N} \left( \boldsymbol{\mu}_{\theta,t}, \text{diag}(\boldsymbol{\sigma}_{\theta,t}^2) \right), \quad \left[ \boldsymbol{\mu}_{\theta,t}, \boldsymbol{\sigma}_{\theta,t}^2 \right] = \theta^{\text{prior}}(\boldsymbol{d}_{t-1}, \boldsymbol{a}_{t-1}),$$

$$\boldsymbol{x}_t|\boldsymbol{z}_t \sim \mathcal{N} \left( \boldsymbol{\mu}_{x,t}, \text{diag}(\boldsymbol{\sigma}_{x,t}^2) \right), \quad \left[ \boldsymbol{\mu}_{x,t}, \boldsymbol{\sigma}_{x,t}^2 \right] = \theta^{\text{decoder}}(\boldsymbol{z}_t, \boldsymbol{d}_{t-1}). \quad (10)$$

For building recurrent connections, the choice of RNN types is not limited. In our study, the *long-short term memory* (LSTM) (Hochreiter & Schmidhuber, 1997) is used since it works well in general cases. So we have $\boldsymbol{d}_t = \text{LSTM}(\boldsymbol{d}_{t-1}; \boldsymbol{z}_t, \boldsymbol{x}_t)$.

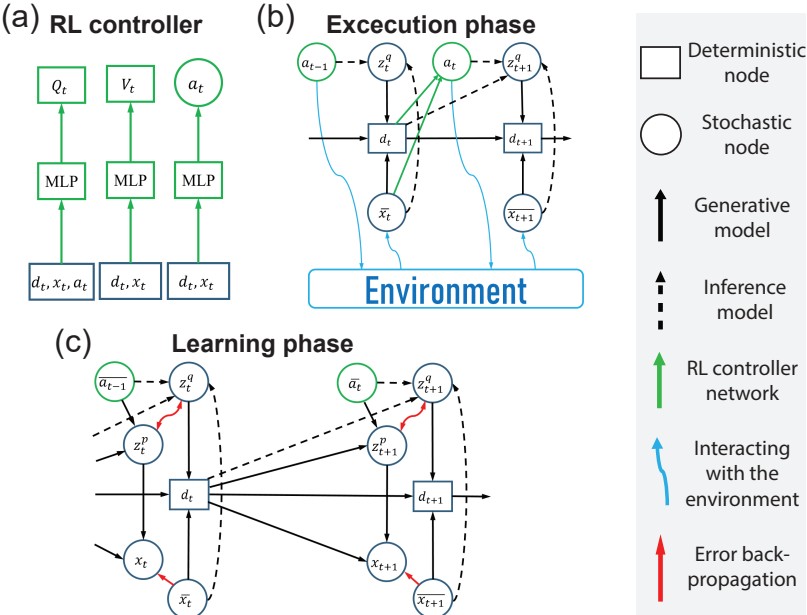

Figure 2: Computation graph of the proposed algorithm. **(a)** The RL controller. **(b)** The execution phase. **(c)** The learning phase of a VRM. $\boldsymbol{a}$: action; $\boldsymbol{z}$: latent variable; $\boldsymbol{d}$: RNN state variable; $\boldsymbol{x}$: raw observation (including reward); $Q$: state-action value function; $V$: state value function. A bar on a variable means that it is the actual value from the replay buffer or the environment. Each stochastic variable follows a parameterized diagonal Gaussian distribution.

As in training a VRNN, the VRM is trained by maximizing an evidence lower bound (Fig. 1(c))

$$ELBO = \sum_t \left\{ \mathbb{E}_{q_\phi} \left[ \log p_\theta(\boldsymbol{x}_t | \boldsymbol{z}_{1:t}, \boldsymbol{x}_{1:t-1}) \right] \right.$$
$$\left. - D_{KL} \left[ q_\phi(\boldsymbol{z}_t | \boldsymbol{z}_{1:t-1}, \bar{\boldsymbol{x}}_{1:t}, \bar{\boldsymbol{a}}_{1:t}) || p_\theta(\boldsymbol{z}_t | \boldsymbol{z}_{1:t-1}, \bar{\boldsymbol{x}}_{1:t-1}, \bar{\boldsymbol{a}}_{1:t}) \right] \right\}. \tag{11}$$

In practice, the first term $\mathbb{E}_{q_\phi} \left[ \log p_\theta(\boldsymbol{x}_t | \boldsymbol{z}_{1:t}, \boldsymbol{x}_{1:t-1}) \right]$ can be obtained by unrolling the RNN using the inference model (Fig. 1(c)) with sampled sequences of $\boldsymbol{x}_t$. Since $q_\phi$ and $p_\theta$ are parameterized Gaussian distributions, the KL-divergence term can be analytically expressed as

$$D_{KL} \left[ q_\phi(\boldsymbol{z}_t) || p_\theta(\boldsymbol{z}_t) \right] = \log \frac{\boldsymbol{\sigma}_{\phi,t}}{\boldsymbol{\sigma}_{\theta,t}} + \frac{(\boldsymbol{\mu}_{\phi,t} - \boldsymbol{\mu}_{\theta,t})^2 + \boldsymbol{\sigma}_{\phi,t}^2}{2\boldsymbol{\sigma}_{\theta,t}^2} - \frac{1}{2} \tag{12}$$

For computation efficiency in experience replay, we train a VRM by sampling minibatchs of truncated sequences of fixed length, instead of whole episodes. Details are found in Appendix A.1.

Since training of a VRM is segregated from training of the RL controllers, there are several strategies for conducting them in parallel. For the RL controller, we adopted a smooth update strategy as in Haarnoja et al. (2018a), i.e., performing one time of experience replay every $n$ steps. To train the VRM, one can also conduct smooth update. However, in that case, RL suffers from instability of the representation of underlying states in the VRM before it converges. Also, stochasticity of RNN state variables $\boldsymbol{d}$ can be meaninglessly high at early stage of training, which may create problems in RL. Another strategy is to pre-train the VRM for abundant epochs only before RL starts, which unfortunately, can fail if novel observations from the environment appear after some degree of policy improvement. Moreover, if pre-training and smooth update are both applied to the VRM, RL may suffer from a large representation shift of the belief state.

To resolve this conflict, we propose using two VRMs, which we call the *first-impression model* and the *keep-learning model*, respectively. As the names suggest, we pre-train the first-impression model and stop updating it when RL controllers and the keep-learning model start smooth updates. Then we take state variables from both VRMs, together with raw observations, as input for the RL controller. We found that this method yields better overall performance than using a single VRM (Appendix C).

---

**Algorithm 1 Variational Recurrent Models with Soft Actor Critic**

---

Initialize the first-impression VRM $\mathcal{M}_f$ and the keep-learning VRM $\mathcal{M}_k$, the RL controller $\mathcal{C}$, and the replay buffer $\mathcal{D}$, global step $t \leftarrow 0$.
**repeat**
    Initialize an episode, assign $\mathcal{M}$ with zero initial states.
    **while** episode not terminated **do**
        Sample an action $\boldsymbol{a}_t$ from $\pi(\boldsymbol{a}_t | \boldsymbol{d}_t, \boldsymbol{x}_t)$ and execute $a_t$, $t \leftarrow t + 1$.
        Record $(\boldsymbol{x}_t, \boldsymbol{a}_t, done_t)$ into $\mathcal{B}$.
        Compute 1-step forward of both VRMs using inference models.
        **if** $t == step\_start\_RL$ **then**
            For $N$ epochs, sample a minibatch of samples from $\mathcal{B}$ to update $\mathcal{M}_f$ (Eq. 11).
        **end if**
        **if** $t > step\_start\_RL$ and $mod(t, train\_interval\_KLVRM) == 0$ **then**
            Sample a minibatch of samples from $\mathcal{B}$ to update $\mathcal{M}_k$ (Eq. 5, 6, 7, 8) .
        **end if**
        **if** $t > step\_start\_RL$ and $mod(t, train\_interval\_RL) == 0$ **then**
            Sample a minibatch of samples from $\mathcal{B}$ to update $\mathcal{R}$ (Eq. 11) .
        **end if**
    **end while**
**until** training stopped

---

### 4.2 REINFORCEMENT LEARNING CONTROLLERS

As shown in Fig. 1(a), we use multi-layer perceptrons (MLP) as function approximators for $V$, $Q$, respectively. Inputs for the $Q_t$ network are $(\boldsymbol{x}_t, \boldsymbol{d}_t, \boldsymbol{a}_t)$, and $V_t$ is mapped from $(\boldsymbol{x}_t, \boldsymbol{d}_t)$. Following Haarnoja et al. (2018a), we use two Q networks $\lambda_1$ and $\lambda_2$ and compute $Q = \min(Q_{\lambda_1}, Q_{\lambda_2})$ in Eq. 5 and 7 for better performance and stability. Furthermore, we also used a target value network for computing $V$ in Eq. 6 as in Haarnoja et al. (2018a). The policy function $\pi_\eta$ follows a parameterized Gaussian distribution $\mathcal{N}(\boldsymbol{\mu}_\eta(\boldsymbol{d}_t, \boldsymbol{x}_t), diag(\boldsymbol{\sigma}_\eta(\boldsymbol{d}_t, \boldsymbol{x}_t)))$ where $\boldsymbol{\mu}_\eta$ and $\boldsymbol{\sigma}_\eta$ are also MLPs.

In the execution phase (Fig. 1(b)), observation and reward $\boldsymbol{x}_t = (\boldsymbol{X}_t, r_{t-1})$ are received as VRM inputs to compute internal states $\boldsymbol{d}_t$ using inference models. Then, the agent selects an action, sampled from $\pi_\eta(\boldsymbol{a}_t | \boldsymbol{d}_t, \boldsymbol{x}_t)$, to interact with the environment.

To train RL networks, we first sample sequences of steps from the replay buffer as minibatches; thus, $\boldsymbol{d}_t$ can be computed by the inference models using recorded observations $\bar{\boldsymbol{x}}_t$ and actions $\bar{\boldsymbol{a}}_t$ (See Appendix A.1.2). Then RL networks are updated by minimizing the loss functions with gradient descent. Gradients stop at $\boldsymbol{d}_t$ so that training of RL networks does not involve updating VRMs.

## 5 RESULTS

To empirically evaluate our algorithm, we performed experiments in a range of (partially observable) continuous control tasks and compared it to the following alternative algorithms. The overall procedure is summarized in Algorithm 1. For the RL controllers, we adopted hyperparameters from the original SAC implementation (Haarnoja et al., 2018b). Both the keep-learning and first-impression VRMs were trained using learning rate 0.0008. We pre-trained the first-impression VRM for 5,000 epochs, and updated the keep-learning VRM every 5 steps. Batches of size 4, each containing a sequence of 64 steps, were used for training both the VRMs and the RL controllers. All tasks used the same hyperparameters (Appendix A.1).

- **SAC-MLP**: The vanilla soft actor-critic implementation (Haarnoja et al., 2018a;b), in which each function is approximated by a 2-layer MLP taking raw observations as input.
- **SAC-LSTM**: Soft actor-critic with recurrent networks as function approximators, where raw observations are processed through an LSTM layer followed by 2 layers of MLPs. This allows the agent to make decisions based on the whole history of raw observations. In this case, the network has to conduct representation learning and dynamic programming collectively. Our algorithm is compared with SAC-LSTM to demonstrate the effect of separating representation learning from dynamic programming.

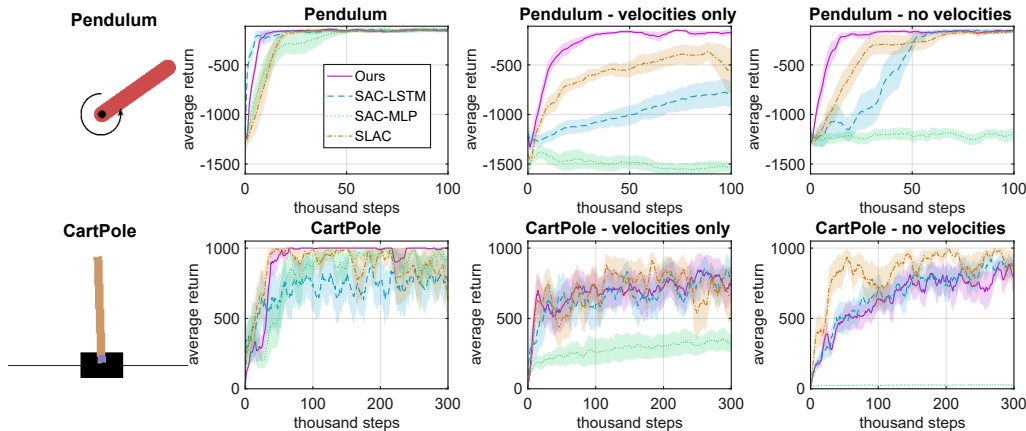

Figure 3: Learning curves of the classic control tasks. Shaded areas indicate S.E.M..

- **SLAC**: The stochastic latent actor-critic algorithm introduced in Lee et al. (2019), which is a state-of-the-art RL algorithm for solving POMDP tasks. It was shown that SLAC outperformed other model-based and model-free algorithms, such as (Igl et al., 2018; Hafner et al., 2018), in robotic control tasks with third-person image of the robot as observation[2].

Note that in our algorithm, we apply pre-training of the first-impression model. For a fair comparison, we also perform pre-training for the alternative algorithm with the same epochs. For SAC-MLP and SAC-LSTM, pre-training is conducted on RL networks; while for SLAC, its model is pre-trained.

## 5.1 PARTIALLY OBSERVABLE CLASSIC CONTROL TASKS

The *Pendlum* and *CartPole* (Barto et al., 1983) tasks are the classic control tasks for evaluating RL algorithms (Fig. 3, Left). The CartPole task requires learning of a policy that prevents the pole from falling down and keeps the cart from running away by applying a (1-dimensional) force to the cart, in which observable information is the coordinate of the cart, the angle of the pole, and their derivatives w.r.t time (i.e., velocities). For the Pendulum task, the agent needs to learn a policy to swing an inverse-pendulum up and to maintain it at the highest position in order to obtain more rewards.

We are interested in classic control tasks because they are relatively easy to solve when fully observable, and thus the PO cases can highlight the representation learning problem. Experiments were performed in these two tasks, as well as their PO versions, in which either velocities cannot be observed or only velocities can be observed. The latter case is meaningful in real-life applications because an agent may not be able to perceive its own position, but can estimate its speed.

As expected, SAC-MLP failed to solve the PO tasks (Fig. 3). While our algorithm succeeded in learning to solve all these tasks, SAC-LSTM showed poorer performance in some of them. In particular, in the pendulum task with only angular velocity observable, SAC-LSTM may suffer from the periodicity of the angle. SLAC performed well in the CartPole tasks, but showed less satisfactory sample efficiency in the Pendulum tasks.

## 5.2 PARTIALLY OBSERVABLE ROBOTIC CONTROL TASKS

To examine performance of the proposed algorithm in more challenging control tasks with higher degrees of freedom (DOF), we also evaluated performance of the proposed algorithm in the OpenAI *Roboschool* environments (Brockman et al., 2016). The Roboschool environments include a number

---

[2]SLAC was developed for pixel observations. To compare it with our algorithm, we made some modifications of its implementation (see Appendix A.2.3). Nonetheless, we expect the comparison can demonstrate the effect of the key differences as aforementioned in Section 2.

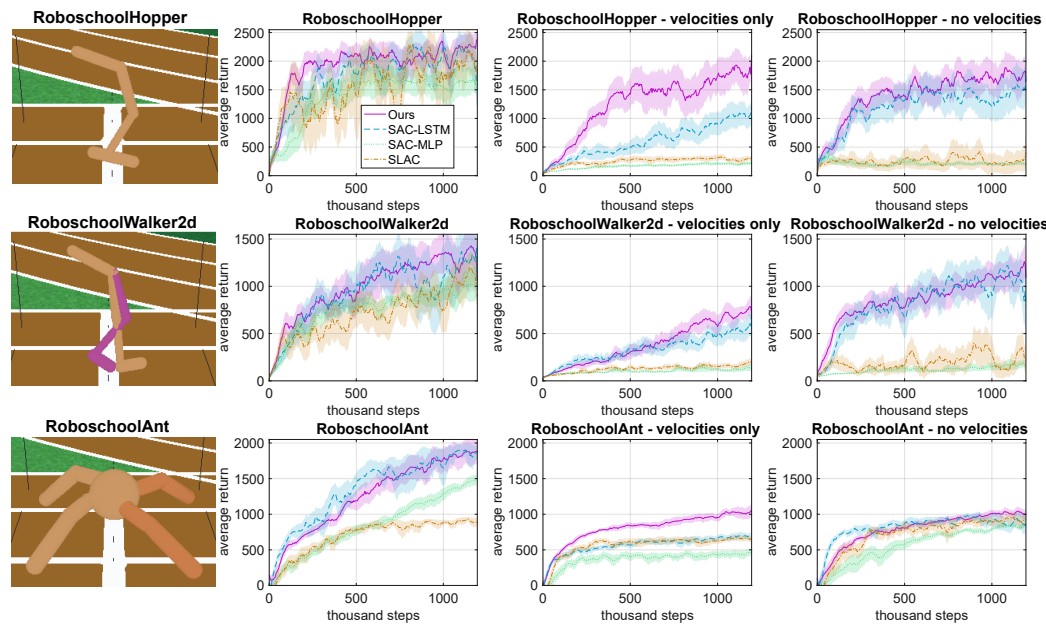

Figure 4: Learning curves of the robotic control tasks, plotted in the same way as in Fig. 3.

of continuous robotic control tasks, such as teaching a multiple-joint robot to walk as fast as possible without falling down (Fig. 4, Left). The original Roboschool environments are nearly fully observable since observations include the robot's coordinates and (trigonometric functions of) joint angles, as well as (angular and coordinate) velocities. As in the PO classic control tasks, we also performed experiments in the PO versions of the Roboschool environments.

Using our algorithm, experimental results (Fig. 4) demonstrated substantial policy improvement in all PO tasks (visualization of the trained agents is in Appendix D). In some PO cases, the agents achieved comparable performance to that in fully observable cases. For tasks with unobserved velocities, our algorithm performed similarly to SAC-LSTM. This is because velocities can be simply estimated by one-step differences in robot coordinates and joint angles, which eases representation learning. However, in environments where only velocities can be observed, our algorithm significantly outperformed SAC-LSTM, presumably because SAC-LSTM is less efficient at encoding underlying states from velocity observations. Also, we found that learning of a SLAC agent was unstable, i.e., it sometimes could acquire a near-optimal policy, but often its policy converged to a poor one. Thus, average performance of SLAC was less promising than ours in most of the PO robotic control tasks.

## 5.3 LONG-TERM MEMORIZATION TASKS

Another common type of PO task requires long-term memorization of past events. To solve these tasks, an agent needs to learn to extract and to remember critical information from the whole history of raw observations. Therefore, we also examined our algorithm and other alternatives in a long-term memorization task known as the *sequential target reaching task* (Han et al., 2019), in which a robot agent needs to reach 3 different targets in a certain sequence (Fig. 5, Left). The robot can control its two wheels to move or turn, and will get one-step small, medium, and large rewards when it reaches the first, second, and third targets, respectively, in the correct sequence. The robot senses distances and angles from the 3 targets, but does not receive any signal indicating which target to reach. In each episode, the robot's initial position and those of the three targets are randomly initialized. In order to obtain rewards, the agent needs to infer the current correct target using historical observations.

We found that agents using our algorithm achieved almost 100% success rate (reaching 3 targets in the correct sequence within maximum steps). SAC-LSTM also achieved similar success rate after convergence, but spent more training steps learning to encode underlying goal-related information

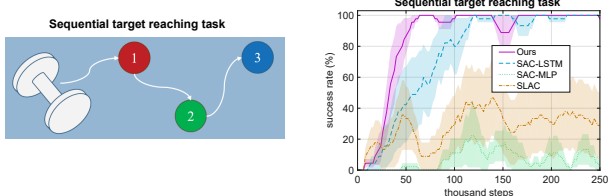

Figure 5: Learning curves of the sequential target reaching task.

from sequential observations. Also, SLAC struggled hard to solve this task since its actor only received a limited steps of observations, making it difficult to infer the correct target.

### 5.4 CONVERGENCE OF THE KEEP-LEARNING VRM

One of the most concerned problems of our algorithm is that input of the RL controllers can experience representation change, because the keep-learning model is not guaranteed to converge if novel observation appears due to improved policy (e.g. for a hopper robot, "in-the-air" state can only happen after it learns to hop). To empirically investigate how convergence of the keep-learning VRM affect policy improvement, we plot the loss functions (negative ELBOs) of the the keep-learning VRM for 3 example tasks (Fig. 6). For a simpler task (CartPole), the policy was already near optimal before the VRM fully converged. We also saw that the policy was gradually improved after the VRM mostly converged (RoboschoolAnt - no velocities), and that the policy and the VRM were being improved in parallel (RoboschoolAnt - velocities only).

The results suggested that policy could be improved with sufficient sample efficiency even the keep-learning VRM did not converge. This can be explained by that the RL controller also extract information from the first-impression model and the raw observations, which did not experience representation change during RL. Indeed, our ablation study showed performance degradation in many tasks without the first-impression VRM (Appendix C).

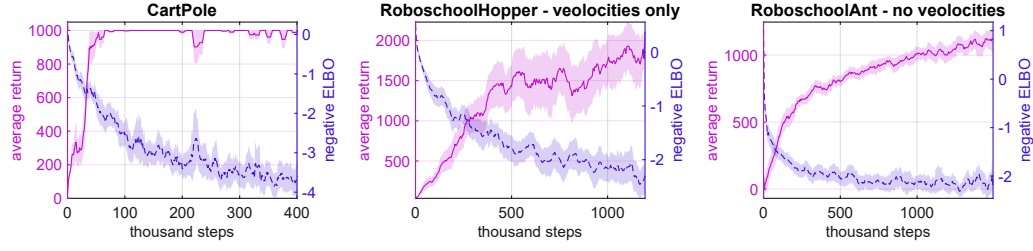

Figure 6: Example tasks showing relationship between average return of the agent and negative ELBO (loss function, dashed) of the keep-learning VRM.

## 6 DISCUSSION

In this paper, we proposed a variational recurrent model for learning to represent underlying states of PO environments and the corresponding algorithm for solving POMDPs. Our experimental results demonstrate effectiveness of the proposed algorithm in tasks in which underlying states cannot be simply inferred using a short sequence of observations. Our work can be considered an attempt to understand how RL benefits from stochastic Bayesian inference of state-transitions, which actually happens in the brain (Funamizu et al., 2016), but has been considered less often in RL studies.

We used stochastic models in this work which we actually found perform better than deterministic ones, even through the environments we used are deterministic (Appendix C). The VRNN can

be replaced with other alternatives (Bayer & Osendorfer, 2014; Goyal et al., 2017) to potentially improve performance, although developing model architecture is beyond the scope of the current study. Moreover, a recent study (Ahmadi & Tani, 2019) showed a novel way of inference using back-propagation of prediction errors, which may also benefit our future studies.

Many researchers think that there are two distinct systems for model-based and model-free RL in the brain (Gläscher et al., 2010; Lee et al., 2014) and a number of studies investigated how and when the brain switches between them (Smittenaar et al., 2013; Lee et al., 2014). However, Stachenfeld et al. (2017) suggested that the hippocampus can learn a *successor representation* of the environment that benefits both model-free and model-based RL, contrary to the aforementioned conventional view. We further propose another possibility, that a model is learned, but not used for planning or dreaming. This blurs the distinction between model-based and model-free RL.

## ACKNOWLEDGEMENT

This work was supported by Okinawa Institute of Science and Technology Graduate University funding, and was also partially supported by a Grant-in-Aid for Scientific Research on Innovative Areas: Elucidation of the Mathematical Basis and Neural Mechanisms of Multi-layer Representation Learning 16H06563. We would like to thank the lab members in the Cognitive Neurorobotics Research Unit and the Neural Computation Unit of Okinawa Institute of Science and Technology. In particular, we would like to thank Ahmadreza Ahmadi for his help during model development. We also would like to thank Steven Aird for assisting improving the manuscript.

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

## A    Implementation Details

In this section we describe the details of implementing our algorithm as well as the alternative ones. Summaries of hyperparameters can be found in Table 1 and 2.

Table 1: Shared hyperparameters for all the algorithms and tasks in the paper, adopted from the original SAC implementation (Haarnoja et al., 2018b).

| Hyperparameter | Description | Value |
| --- | --- | --- |
| $\gamma$ | Discount factor | 0.99 |
| step_start_RL | From how many steps to start training the RL controllers | 1,000 |
| train_interval_RL | Interval of training the RL controllers | 1 |
| lr_actor | Learning rate for the actor | 0.0003 |
| lr_critic | Learning rate for the critic | 0.0003 |
| lr_$\alpha$ | Learning rate for the entropy coefficient $\alpha$ | 0.0003 |
| $\mathcal{H}_{\text{tar}}$ | Target entropy | $-$DOF |
| optimizer | Optimizers for all the networks | Adam (Kingma & Ba, 2014) |
| $\tau$ | Fraction of updating the target network each gradient step | 0.005 |
| policy_layers | MLP layer sizes for $\boldsymbol{\mu}_\eta$ and $\boldsymbol{\pi}_\eta$ | 256, 256 |
| value_layers | MLP layer sizes for $V_\phi$ and $Q_\lambda$ | 256, 256 |

Table 2: Hyperparameters for the proposed algorithm.

| Hyperparameter | Description | Value |
| --- | --- | --- |
| train_times_FIVRM | Epoches of training the first-impression model. | 5,000 |
| train_interval_KLVRM | Interval of training the keep-learning model. | 5 |
| lr_model | Learning rate for the VRMs | 0.0008 |
| seq_len | How many steps in a sampled sequence for each update | 64 |
| batch_size | How many sequences to sample for each update | 4 |

### A.1    The Proposed Algorithm

#### A.1.1    Network architectures

The first-impression model and the keep-learning model adopted the same architecture. Size of $\boldsymbol{d}$ and $\boldsymbol{z}$ is 256 and 64, respectively. We used one-hidden-layer fully-connected networks with 128 hidden neurons for the inference models $\left[\boldsymbol{\mu}_{\phi,t}, \boldsymbol{\sigma}_{\phi,t}^2\right] = \phi(\boldsymbol{x}_t, \boldsymbol{d}_{t-1}, \boldsymbol{a}_{t-1})$, as well as for $\left[\boldsymbol{\mu}_{\theta,t}, \boldsymbol{\sigma}_{\theta,t}^2\right] = \theta^{\text{prior}}(\boldsymbol{d}_{t-1}, \boldsymbol{a}_{t-1})$ in the generative models. For the decoder $\left[\boldsymbol{\mu}_{x,t}, \boldsymbol{\sigma}_{x,t}^2\right] = \theta^{\text{decoder}}(\boldsymbol{z}_t, \boldsymbol{d}_{t-1})$ in the generative models, we used 2-layers MLPs with 128 neurons in each layer. The input processing layer $f_x$ is also an one-layer MLP with size-128. For all the Gaussian variables, output functions for mean are linear and output functions for variance are softplus. Other activation functions of the VRMs are tanh.

The RL controllers are the same as those in SAC-MLP (Section A.2.1) except that network inputs are raw observations together with the RNN states from the first-impression model and the keep-learning model.

#### A.1.2    Initial states of the VRMs

To train the VRMs, one can use a number of entire episodes as a mini-batch, using zero initial states, as in Heess et al. (2015). However, when tackling with long episodes (e.g. there can be 1,000 steps in each episode in the robotic control tasks we used) or even infinite-horizon problems, the computation consumption will be huge in back-propagation through time (BPTT). For better computation efficiency, we used 4 length-64 sequences for training the RNNs, and applied the *burn-in* method for providing the initial states (Kapturowski et al., 2018), or more specifically, unrolling the RNNs using a portion of the replay sequence (burn-in period, up to 64 steps in our case) from zero

initial states. We assume that proper initial states can be obtained in this way. This is crucial for the tasks that require long-term memorization, and is helpful to reduce bias introduces by incorrect initial states in general cases.

## A.2 ALTERNATIVE ALGORITHMS

### A.2.1 SAC-MLP

We followed the original implementation of SAC in (Haarnoja et al., 2018a) including hyperparameters. However, we also applied automatic learning of the entropy coefficient $\alpha$ (inverse of the the reward scale in Haarnoja et al. (2018a)) as introduced by the authors in Haarnoja et al. (2018b) to avoid tuning the reward scale for each task.

### A.2.2 SAC-LSTM

To apply recurrency to SAC's function approximators, we added an LSTM network with size-256 receiving raw observations as input. The function approximators of actor and critic were the same as those in SAC except receiving the LSTM's output as input. The gradients can pass through the LSTM so that the training of the LSTM and MLPs were synchronized. The training the network also followed Section A.1.2.

### A.2.3 SLAC

We mostly followed the implementation of SLAC explained in the authors' paper (Lee et al., 2019). One modification is that since their work was using pixels as observations, convolutional neural networks (CNN) and transposed CNNs were chosen for input feature extracting and output decoding layers; in our case, we replaced the CNN and transposed CNNs by 2-layers MLPs with 256 units in each layer. In addition, the authors set the output variance $\sigma_{y,t}^2$ for each image pixel as 0.1. However, $\sigma_{y,t}^2 = 0.1$ can be too large for joint states/velocities as observations. We found that it will lead to better performance by setting $\boldsymbol{\sigma}_{y,t}$ as trainable parameters (as that in our algorithm). We also used a 2-layer MLP with 256 units for approximating $\boldsymbol{\sigma}_y(\boldsymbol{x}_t, \boldsymbol{d}_{t-1})$. To avoid network weights being divergent, all the activation functions of the model were tanh except those for outputs.

## B ENVIRONMENTS

For the robotic control tasks and the Pendulum task, we used environments (and modified them for PO versions) from OpenAI Gym (Brockman et al., 2016). The CartPole environment with a continuous action space was from Danforth (2018), and the codes for the sequential target reaching tasks were provided by the authors (Han et al., 2019).

In the no-velocities cases, velocity information was removed from raw observations; while in the velocities-only cases, only velocity information was retained in raw observations. We summarize key information of each environment in Table 3.

The performance curves were obtained in evaluation phases in which agents used same policy but did not update networks or record state-transition data. Each experiment was repeated using 5 different random seeds.

## C ABLATION STUDY

This section demonstrated a ablation study in which we compared the performance of the proposed algorithm to the same but with some modification:

- **With a single VRM**. In this case, we used only one VRM and applied both pre-training and smooth update to it.

- **Only first-impression model**. In this case, only the first-impression model was used and pre-trained.

Table 3: Information of environments we used.

| Name | dim($\mathbb{X}$) | DOF | Maximum steps |
|------|------|-----|---------------|
| Pendulum | 3 | 1 | 200 |
| Pendulum (velocities only) | 1 | 1 | 200 |
| Pendulum (no velocities) | 2 | 1 | 200 |
| CartPole | 4 | 1 | 1,000 |
| CartPole (velocities only) | 2 | 1 | 1,000 |
| CartPole (no velocities) | 2 | 1 | 1,000 |
| RoboschoolHopper | 15 | 3 | 1,000 |
| RoboschoolHopper (velocities only) | 6 | 3 | 1,000 |
| RoboschoolHopper (no velocities) | 9 | 3 | 1,000 |
| RoboschoolWalker2d | 22 | 6 | 1,000 |
| RoboschoolWalker2d (velocities only) | 9 | 6 | 1,000 |
| RoboschoolWalker2d (no velocities) | 13 | 6 | 1,000 |
| RoboschoolAnt | 28 | 8 | 1,000 |
| RoboschoolAnt (velocities only) | 11 | 8 | 1,000 |
| RoboschoolAnt (no velocities) | 17 | 8 | 1,000 |
| Sequential goal reaching task | 12 | 2 | 128 |

- **Only keep-learning model**. In this case, only the keep-learning model was used and smooth-update was applied.
- **Deterministic model**. In this case, the first-imporession model and the keep-learning model were deterministic RNNs which learned to model the state-transitions by minimizing mean-square error between prediction and observations instead of $ELBO$. The network architecture was mostly the same as the VRM expect that the inference model and the generative model were merged into a deterministic one.

The learning curves are shown in Fig. 7. It can be seen that the proposed algorithm consistently performed similar as or better than the modified ones.

## D    VISUALIZATION OF TRAINED AGENTS

Here we show actual movements of the trained robots in the PO robotic control tasks (Fig. 8). It can be seen that the robots succeeded in learning to hop or walk, although their policy may be sub-optimal.

## E    MODEL ACCURACY

As we discussed in Section 2, our algorithm relies mostly on encoding capacity of models, but does not require models to make accurate prediction of future observations. Fig. 9 shows open-loop (using the inference model to compute the latent variable $z$) and close-loop (purely using the generative model) prediction of raw observation by the keep-learning models of randomly selected trained agents. Here we showcase "RoboschoolHopper - velocities only" and "Pendulum - no velocities" because in these tasks our algorithm achieved similar performance to those in fully-observable versions (Fig. 4), although the prediction accuracy of the models was imperfect.

## F    SENSITIVITY TO HYPERPARAMETERS OF THE VRMS

To empirically show how choice of hyperparameters of the VRMs affect RL performance, we conducted experiments using hyperparameters different from those used in the main study. More specifically, the learning rate for both VRMs was randomly selected from {0.0004, 0.0006, 0.0008, 0.001} and the sequence length was randomly selected from {16, 32, 64} (the batch size was $256/(sequence\_length)$ to ensure that the total number of samples in a batch was 256 which matched with the alternative approaches). The other hyperparameters were unchanged.

The results can be checked in Fig 10 for all the environments we used. The overall performance did not significantly change using different, random hyperparameters of the VRMs, although we could observe significant performance improvement (e.g. RoboshoolWalker2d) or degradation (e.g. RoboshoolHopper - velocities only) in a few tasks using different haperparameters. Therefore, the representation learning part (VRMs) of our algorithm does not suffer from high sensitivity to hyperparameters. This can be explained by the fact that we do not use a bootstrapping (e.g. the estimation of targets of value functions depends on the estimation of value functions) (Sutton & Barto, 1998) update rule to train the VRMs.

## G  SCALABILITY

Table 4 showed scalability of our algorithm and the alternative ones.

| Algorithm | wall-clock time (100,000 steps) | # parameters |
|:---:|:---:|:---:|
| Ours | 8 hr | 2.8M |
| SAC-MLP | 1 hr | 0.4M |
| SAC-LSTM | 12 hr | 1.1M |
| SLAC | 5 hr | 2.8M |

Table 4: Wall-clock time and number of parameters of our algorithm and the alternative ones. The working environment was a desktop computer using Intel i7-6850K CPU and the task is "Velocities-only RoboschoolHopper". The wall-clock time include training the first-impression VRM or pre-trainings.

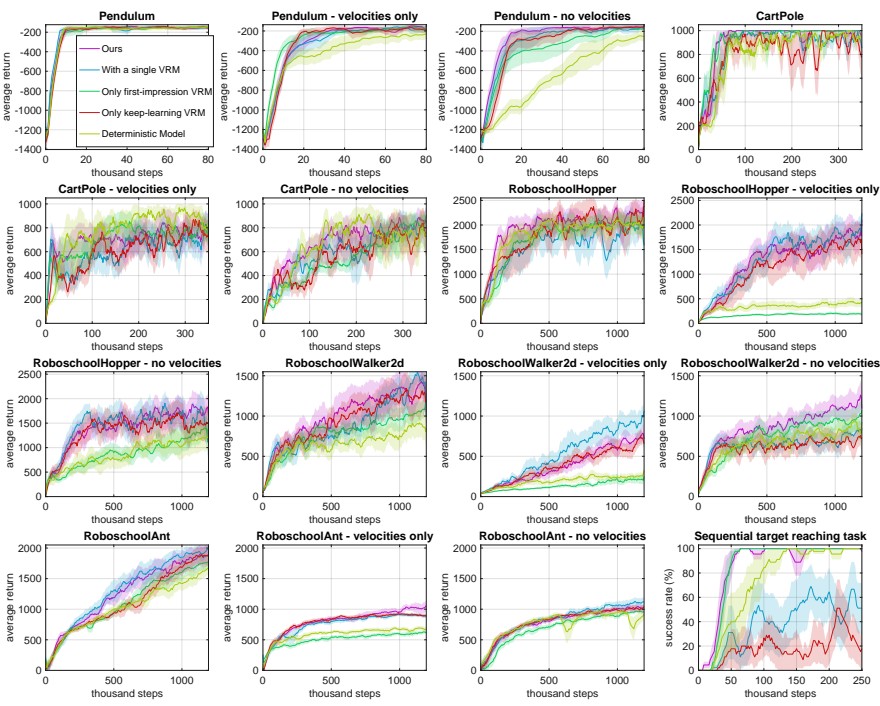

Figure 7: Learning curves of our algorithms and the modified ones.

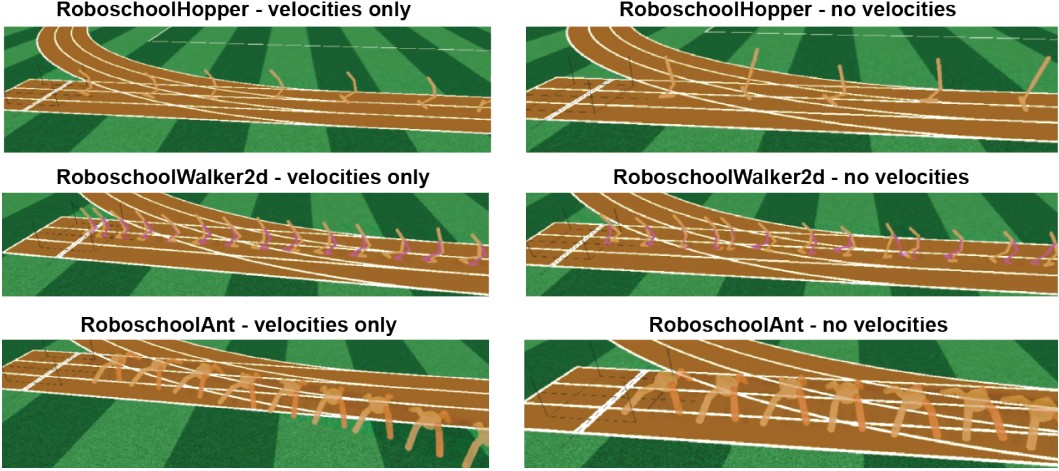

Figure 8: Robots learned to hop or walk in PO environments using our algorithm. Each panel shows trajectory of a trained agent (randomly selected) within one episode.

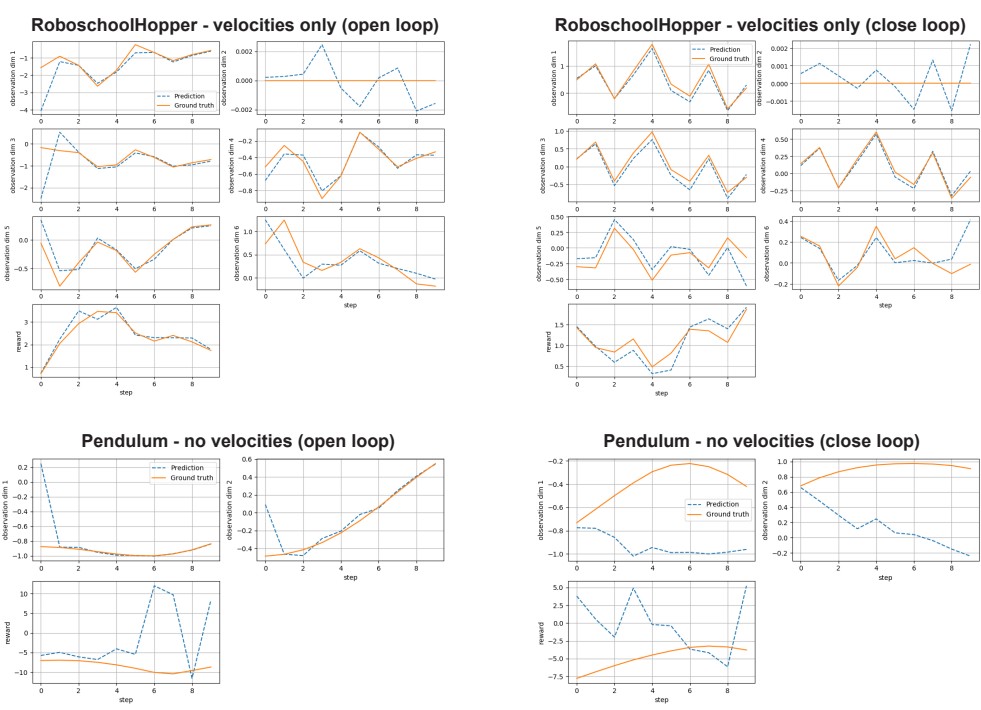

Figure 9: Examples of observation predictions by keep-learning VRMs of trained agents.

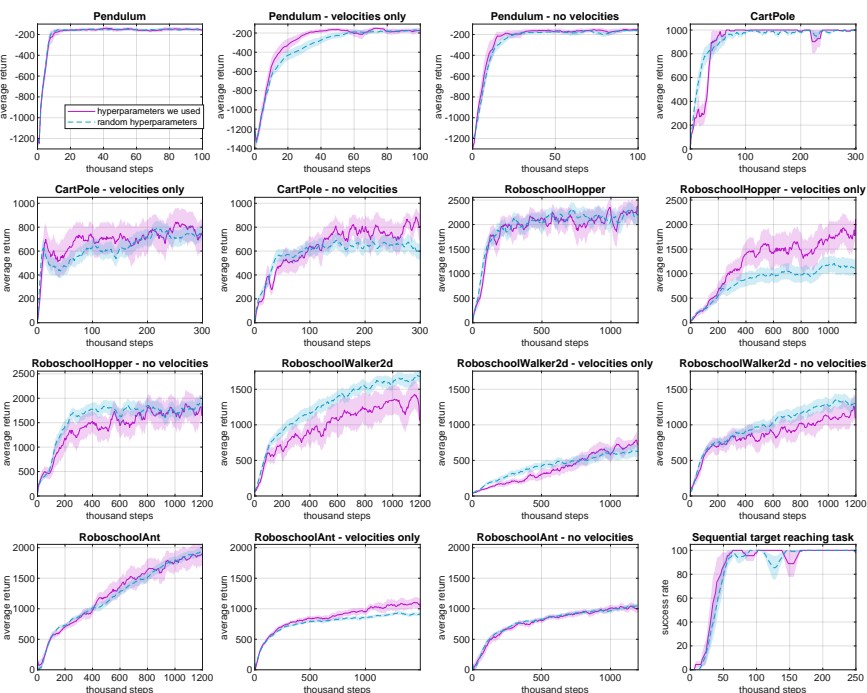

Figure 10: The learning curves of our algorithm using the hyperparameters for the VRMs used in the paper (Table 2), and using a range of random hyperparameters (Appendix F). Data are Mean $\pm$ S.E.M., obtained from 20 repeats using different random seeds.

