# OpenReview forum: "Variational Recurrent Models for Solving Partially Observable Control Tasks"
_ICLR.cc/2020/Conference — Accept (Poster)_

### Official Review · AnonReviewer2 · 2019-10-23
**Official Blind Review #2**

**Rating:** 8

**Review:**

* Summary

The work considers reinforcement learning in partially observed environments, targeting in particular the case when the agent needs to remember some important information over a longer period of time.

The work gives an interesting connection between variational recurrent neural networks and reinforcement learning. The paper is well written and technically sound. The method achieves improved performance in several (simpler) benchmark problems in comparison to strong baselines.

* Detailed discussion

The work allocates latent variables to describe unobserved part of the state and learns the state transition probability model (the action-dependent generator (10)). Learning these state transitions becomes an unsupervised learning task, for which the lower bound on the likelihood of the observed data is optimized. This learning of the state transition model is actually similar to classical model-based RL. The policy and value functions are then learned on top of this model using SoTA techniques. From this perspective I would say this is a classical approach to partially observed reinforcement learning with advanced models for transition matrix and value and policy functions.

The work combines together variational recurrent neural network (VRNN) and soft actor critic (SAC) models into a rather advanced system for reinforcement learning. These parts are somewhat extended to fit together, but it is not clear whether all components are needed. For example, why z and d need to be distinguished, cannot they be combined into one latent variable “zd” in order to simplify the number of connections? Or I am missing some important property necessary for the inference?

In (7),(8) are the expectations over the same state / actions or different? The notation is not clear about it.

After (2), given x_t and d_{t-1} instead of d_t?

The experiments are conducted in the setting that the observations were all relevant measurements for the agent (positions, velocities). From Fig 7 RoboscoolHooper (close loop) we see that actually all measurements can be well predicted by generative model up to 8 steps ahead (in dim 2 that stays zero the noise is amplified). This is a nice and desirable property and is the consequence of the generative modeling and the marginal likelihood ELBO optimization. However the paper seem to argue (Appendix E) that the algorithm does not require the model to make accurate predictions and relies on the encoding capacity, which I find confusing. Furthermore, it would be interesting to see how this approach should scale to the setting when there are more sensory inputs, some less relevant than the other. In the end, is a good generative model to predict the future observation needed or not needed in this approach?

The exploration seems to be addressed just by randomizing the policy,  are there any better options?


**Experience Assessment:**

I do not know much about this area.

**Review Assessment: Checking Correctness Of Derivations And Theory:**

I assessed the sensibility of the derivations and theory.

**Review Assessment: Checking Correctness Of Experiments:**

I assessed the sensibility of the experiments.

**Review Assessment: Thoroughness In Paper Reading:**

I read the paper at least twice and used my best judgement in assessing the paper.

---

> ### Author Response · Authors · 2019-11-09
> **Response to reviewer  #2**
>
>
> Thanks your comments and questions, which will be helpful to improve the clarity of our paper!  We here discuss about them as follows.
>
>
> >>> It is not clear whether all components are needed. For example, why z and d need to be distinguished?
>
> We adopted the network design mostly from the original VRNN implementation [Chung 2015]. except that we added an action feedback connection to address the effect of action on state-transition. This “distinguished z and d” setting was also suggested and used by the original VRNN [Chung 2015] paper and many other related works [Shabanian 2017].  The reason is because we need abundant #neurons for expressive power, but we do not want #parameters to be too large. Since z is stochastic variable, there will be much more #parameters (like $\mu_z, \sigma_z$) if the dimension of z is large, but we actually do not need that many of stochastic units. So “more d and less z” appears as a solution considering this tradeoff.
>
>
> >>> In (7),(8) are the expectations over the same state / actions or different? The notation is not clear about it.
>
> Thanks for pointing out this confusion. For (5)~(8), $s_t$ is the same, sampled from the replay buffer.
> For (7), action is reparametrized (indicated by $a_\eta$) to compute the gradients.  This can result in a lower variance estimator, according to the SAC paper [Haarnoja 2018]
> But for (8), no gradient is needed on action (indicated by $a$), so action is just treated as a constant.
> We’ve added explanations to clarify this in the revised version of the paper (you can download from the current webpage)
>
> >>> After (2), given $x_t$ and $d_{t-1}$ instead of $d_t$?
>
> Do you mean $[\mu_{z,t}, \sigma^2_{z,t}] = f^{encoder} (x_t, d_{t-1})$ ? This is because $z_t$ needs to be computed before $d_t$ in the inference model, which is the default design of VRNN.
>
>
> >>> … In the end, is a good generative model to predict the future observation needed or not needed in this approach?
>
> Let us consider the “Pendulum-no velocities” task: Suppose the model cannot accurately predict future observation, but its hidden states can infer whether the pendulum is moving clockwise or counter-clockwise. In this case, the RL controller should be able to acquire a near-optimal policy. Therefore, we consider that a good model is needed to [encode critical information of the environment for control] rather than [to accurately predict future observations/rewards], and the former can be achieved by learning to do the latter (maximizing ELBO).
> However, it is hard to evaluated how good the encoding is without decoding. So we could only show Fig.7 (Fig. 9 in the revised version) to partially address this question. It can be seen that the model for Pendulum-no velocities was not satisfying, while this task still could  be well learned.
> Also, we added a section (5.4) in the revised paper to demonstrate that policy can be improved before the keep-learning VRM fully converges. This suggests that while a good generative model to predict future observations is beneficial, sometimes it is not necessary for the purpose of enhancing control skills.
>
>
>
> >>> The exploration seems to be addressed just by randomizing the policy, are there any better options?
>
> In our work, we used the default exploration strategy of SAC and many other RL algorithms for continuous control, in which action follows a (parameterized) Gaussian distribution. We did so since we wanted to focus on discussion the benefits of VRM, but not RL algorithm.
> However, this is a great point. There are a number of interesting studies on a better exploration strategy for RL, such as temporally-correlated noise [Lillicrap 2015], and noisy neural networks for exploration [Fortunato 2017]. Our future study may consider using deterministic node for action, while the stochasticity necessary for exploration is provided by the random variable z. This may lead to a higher-level exploration and it should be interesting to see how it performs.
>
>
>
> References
> -	Chung J, Kastner K, Dinh L, et al. A recurrent latent variable model for sequential data[C]//Advances in neural information processing systems. 2015: 2980-2988.
> -	Shabanian S, Arpit D, Trischler A, et al. Variational bi-lstms[J]. arXiv preprint arXiv:1711.05717, 2017.
> -	Lillicrap T P, Hunt J J, Pritzel A, et al. Continuous control with deep reinforcement learning[J]. arXiv preprint arXiv:1509.02971, 2015.
> -	Fortunato M, Azar M G, Piot B, et al. Noisy networks for exploration[J]. arXiv preprint arXiv:1706.10295, 2017.
> -	Haarnoja T, Zhou A, Hartikainen K, et al. Soft actor-critic algorithms and applications[J]. arXiv preprint arXiv:1812.05905, 2018.

---

### Official Review · AnonReviewer1 · 2019-10-23
**Official Blind Review #1**

**Rating:** 6

**Review:**

The paper proposes that for partially observable reinforcement learning tasks it might be simpler to decompose the problem in two parts: a recurrent world model and a feedforward agent, as opposed to using just a recurrent agent. Intuitively the decomposition makes sense, though it's not very clear to me that the problems encountered training a recurrent agent should be dramatically simpler when training a recurrent world model instead.

For the world model the paper proposes using a variational recurrent state-transition model, which is essentially a VRNN which also conditions on the actions. It is not clear to me that this is easier to learn than a recurrent agent because the probability distribution of the data used to train the modified VRNN is highly dependent on the current policy in the same way that the non-iid training data makes it tricky for the RNN policy models to converge.

While the experimental results seem to show the new algorithm outperforming the existing ones over a large set of random seeds the paper does not specify how the fixed hyperparameters for all models were chosen, leaving open the possibility that different hyperparameter settings would have shown reversals in the experimental results. It's also not clear that the complexity of the alternate models was adequately accounted for (specially since two VRNNs are required to match the performance of a single RNN agent). That said the incompleteness of the experimental results is my only reservation against accepting this paper.


----

The author response somewhat addressed my major concern, which was the lack of hyperparameter tuning in the baselines, as well as clarifying some of the other questions I had. Given that my score has been updated.

**Experience Assessment:**

I have read many papers in this area.

**Review Assessment: Checking Correctness Of Derivations And Theory:**

I assessed the sensibility of the derivations and theory.

**Review Assessment: Checking Correctness Of Experiments:**

I carefully checked the experiments.

**Review Assessment: Thoroughness In Paper Reading:**

I read the paper at least twice and used my best judgement in assessing the paper.

---

> ### Author Response · Authors · 2019-11-09
> **Response to reviewer #1 (Part 1)**
>
> Thanks for pointing out some concerns of our method! Hope the following explanations can address your (and maybe also other readers’) concerns about our claims and results.
>
>
> >>> It's not very clear to me that the problems encountered training a recurrent agent should be dramatically simpler when training a recurrent world model instead. It is not clear to me that this is easier to learn than a recurrent agent because the probability distribution of the data used to train the modified VRNN is highly dependent on the current policy in the same way that the non-iid training data makes it tricky for the RNN policy models to converge.
>
> As you said, the data used to train the variational recurrent model (VRM) depends on the agent’s policy and can be non-iid. However, the VRMs actually are trained by unsupervised learning given the observations and actions. If we assume a VRM is perfectly trained, it should be able to represent the environment regardless the agent’s policy, because state-transitions condition on action, not policy (definition of POMDP).
> Our method is expected to outperform a recurrent agent because most RL algorithms (dynamic programming approaches, such as SAC) are bootstrapping (“All of them update estimates of the values of states based on estimates of the values of successor states. That is, they update estimates on the basis of other estimates.” [Sutton 2018, page 89]). So if a recurrent agent directly learns to represent reward-related state-transitions environments by learning value functions, it may suffer from a biased target due to imperfectly learned value functions. We proposed to solve the representation learning problem by unsupervised learning using the agent’s state-transition data, which is non-bootstrapping and is expected to be easier.
> As for the convergence problem, you are right, it is tricky to for a VRM to converge if policy changes (We guess you mean the VRM by “RNN policy model” since we do not have a RNN policy model.) This is why we proposed two VRMs, the first-impression model (FIM) and the keep-learning model (KLM), as addressed in the last 2 paragraphs in Section 4.1.  The FIM should converge given fixed policy before the RL begins. The KLM is not guaranteed to converge quickly because the agent keeps improve its policy and novel state-transition samples can appear. However, the KLM should also converge after the agent’s policy converged.
> To empirically investigate how convergence of the KLM affect policy improvement, we revised the paper by adding corresponding results and discussions in Section 5.4 (as you can download from the current webpage). We hope the new Section 5.4 can help to address this concern.
>
>
> >>> The paper does not specify how the fixed hyperparameters for all models were chosen, leaving open the possibility that different hyperparameter settings would have shown reversals in the experimental results.
>
> This is actually a good point! It was our mistake not to mention how we selected hyperparameters, which causes concerns because this is indeed important in RL. Let us explain: our algorithm has two parts, the VRMs and the RL controller.
> For the RL controller, we did not tune the hyperparameters. The hyperparameters (Table 1) were exactly the same as used in the original SAC paper [Haarnoja 2018]. Sensitivity to hyperparameters of SAC were discussed in SAC papers. We think it is reasonable to follow their proposed hyperparameters and it is fair to compare with the alternative methods using the same RL hyperparameters.
> For the VRMs, since this representation learning problem is non-bootstrapping, hyperparameters such as learning rate and batch size should not affect too much as long as the ELBO is correctly estimated (of course, too small learning rate will slow down learning). Importantly, we used the same hyperparameters to all the experiments.
> For simplicity, we used the same hyperparameters for both the FIM and KLM (Table 2), which were selected intuitively. We selected a larger learning rate (8e-4 > 3e-4 for RL) and a lower training frequency (0.2 < 1 for RL) for the VRM to save computation cost.
>
> (*Updated Nov 15th*)
> We have performed experiments to investigate the effect of hyperparameters for the VRMs and the results was discussed in Appendix F. The overall performance of our algorithm did not significantly change using different, random hyperparameters of the VRMs (Fig.10 in the revised paper).
>
> We also have added a short explanation of how we selected hyperparameters in the first paragraph of Section 5 of the revised paper.

---

> > ### Author Response · Authors · 2019-11-09
> > **Response to reviewer #1 (Part 2)**
> >
> >
> > >>> It's also not clear that the complexity of the alternate models was adequately accounted for (specially since two VRNNs are required to match the performance of a single RNN agent.
> >
> > It is true that our algorithm uses more parameters than the SAC-LSTM. However, one advantage of our method is that we can use a lower training frequency for the VRMs (at least this works well in our experiments). Since the most time-consuming part is BPTT on RNNs, a lower training frequency for the VRMs can save much time. Indeed, training using our algorithm spends less time to that of SAC-LSTM, though we have more parameters. The following shows the scalability of our approach and alternative ones.
> > For the velocities-only RoboschoolHopper, our algorithm spends around 8 hours for 100k steps (including training the first-impression model) on a desktop computer using an intel i7-6850K CPU.  Our implementation of SLAC need about 5 hours and SAC-LSTM needs 12 hours on the same computer.  SAC-MLP needs only 1 hour since no RNN is trained, given that it mostly failed to learn PO tasks.
> > The total # of parameters is 2.8M for our algorithm (including 2 VRMs and RL controllers), 2.8M for SLAC, 0.4M for SAC, and 1.1 M for SAC-LSTM.
> >
> >
> > References:
> > -	Sutton R S, Barto A G. Reinforcement learning: An introduction[M]. MIT press, 2018.
> > -	Haarnoja T, Zhou A, Hartikainen K, et al. Soft actor-critic algorithms and applications[J]. arXiv preprint arXiv:1812.05905, 2018.

---

> > > ### Comment · AnonReviewer1 · 2019-11-15
> > > **Thanks for the clarifications**
> > >
> > > Thanks for the clarifications!
> > >
> > > I will revise my score upward.

---

### Official Review · AnonReviewer3 · 2019-10-27
**Official Blind Review #3**

**Rating:** 6

**Review:**

Overall the paper is written quite well and addresses a relevant topic. The experiments are thorough and make sense given the research question.  The paper furthermore provide additional ablation studies and does a good job in model analysis.

There are some critical points I have about the paper summarized below:


Introduction:
The 3 categories to solve POMDPs appear artificial. All 3  (whether using a window directly, using a RNN or building a surrogate model) aim to transform the POMDP to an MDP by taking history of observations (either explicit, or implicit).

Figure 1 is too  complicated for an illustrative figure  and the presentation and clarity needs to be improved.


Method:
The idea is sound and described well. However, some design decisions appear ad-hoc  (only justified by empirical observations). For instance, the authors argue it is better to keep 2 models (the "first-impression" and "keep-learning" model) and show an ablation study in appendix C.
Now, one could wonder if the proposed method would still perform better than the baselines if with, say, the just using a single VRM.   The improvement compared to SAC-LSTM is not very large except in a few cases, so possibly just using the VRM would perform no better.

This could then defeat the theoretical argument of the method that "[..] the actor network of SLAC
did not take advantage of the latent variable, but instead used some steps of raw observations as input,
which creates problems in achieving long-term memorization of reward-related state-transitions."


Reference that should be included and possibly compared to:
[1] Watter, Manuel, et al. "Embed to control: A locally linear latent dynamics model for control from raw images." Advances in neural information processing systems. 2015.


Questions:
1. What is the scalability, such as wall-clock time and #parameters  of the approach compared to the baselines? (When using 2 VRMs the # of parameters is doubled)
2. Why the need to input the original observation x (and not just the latent representation) into the RL controller?


**Experience Assessment:**

I have published one or two papers in this area.

**Review Assessment: Checking Correctness Of Derivations And Theory:**

I assessed the sensibility of the derivations and theory.

**Review Assessment: Checking Correctness Of Experiments:**

I assessed the sensibility of the experiments.

**Review Assessment: Thoroughness In Paper Reading:**

I read the paper at least twice and used my best judgement in assessing the paper.

---

> ### Author Response · Authors · 2019-11-09
> **Response to reviewer #3**
>
> Thanks for your review and comments on our work! Many of your points are quite valuable for improving the paper, we respond to them as follows.
>
> >>> The 3 categories to solve POMDPs appear artificial.
>
> To solve POMDPs in which reward relates to history of observations, it is necessary to explicitly or implicitly take history of observations into account. We divide the corresponding methods into 3 categories because they have different implementations which lead to particular pros & cons. Also, we consider the introduction using 3 categories as a quick, short literature review for readers who are not familiar with related studies.
>
>
> >>> Figure 1 is too complicated for an illustrative figure and the presentation and clarity needs to be improved.
>
> Thanks for this valuable suggestion! We have improved its clarity in the revised version (as you can download from the current webpage) by dividing this original Fig. 1 into two. The first one contains Fig.1 (a), and a simplified version of Fig.1(b). The second figure illustrates more detailed graphs as in Fig.1 (b)~(c).
>
>
> >>> However, some design decisions appear ad-hoc. For instance, the authors argue it is better to keep 2 models.
> Now, one could wonder if the proposed method would still perform better than the baselines if with, say, the just using a single VRM.
>
> One of our major design principles is versatility (the algorithm is expected to work on different kinds of tasks without tuning hyper-parameters and model settings).  As we discussed in the last 2 paragraphs of Section 4.1, we worried about the instability of the representation before the VRM converges. This motivated us to conduct for empirical justifications (Appendix C) and we found that the performance using a single VRM in the long-term memorization task was unsatisfactory.
> Nevertheless, we believe that a better design can exists than using 2 VRMs. We currently use this 2-VRMs design because its disadvantage is relatively less obvious (Since we only train the first-impression model before RL starts, the training time is not substantially more than that of using a single model).
>
>
> >>> Reference that should be included and possibly compared to [1] Watter, Manuel, et al
>
> Thanks for the reference. We have cited this work in Section 2 in the revised version. This is an important early work which pioneered in investigating the idea of embedding the environment into latent variables for optimal control. However, one major difference from ours is that they used the model for planning while we do not.
> A fair comparison will be hard because (1) their method is not end-to-end, or more specifically, their hyper-parameters varied on environments. (2) they only modeled 1-step state-transition, making it difficult to capture long-term dependence. (3) their method requires the cost function (reward function) to be given.
>
>
> >>> What is the scalability, such as wall-clock time and #parameters of the approach compared to the baselines? (When using 2 VRMs the # of parameters is doubled)
>
> We are sorry to miss this information in our paper, and we’ve update the paper with the following details (Table 4 in the revised paper).
> For the velocities-only Roboschool Hopper, our algorithm spends around 8 hours for 100k steps (including training the first-impression model) on a desktop computer using an intel i7-6850K CPU.  Our implementation of SLAC need about 5 hours and SAC-LSTM needs 12 hours on the same computer.  SAC-MLP needs only 1 hour since no RNN is trained, given that it mostly failed to learn PO tasks.
> The total # of parameters (including 2 VRMs and RL controllers) is 2.8M for our algorithm, 2.8M for SLAC, 0.4M for SAC, and 1.1 M for SAC-LSTM.
> Note that although our algorithm uses more parameters, the wall-clock time was less than that of SAC-LSTM because we did not train the keep-learning VRM as frequent as training the RL controller. We could use a higher learning rate for a VRM since its learning is not bootstrapping, and train it every 5 steps (RL controller was trained every step).
>
>
> >>> Why the need to input the original observation x (and not just the latent representation) into the RL controller?
>
> The reason is simple: we want our approach to learn efficiently also in fully observable environments, since it is not necessary (but maybe beneficial) to train the VRM(s). Therefore, we tried to input the raw observation x together with RNN states d to the RL controller, and we found that by doing so, our algorithm showed very good sample efficiency also in fully observable cases (such as Pendulum and CartPole in Fig.2).

---

### Author Response · Authors · 2019-11-09
**Short summary of revision**

We thank the reviewers for pointing some shortages of our paper out. We posted a revised version according to the reviews, and here we shortly introduce what has been modified:

1. The original Fig.1 has been divided into two (now Fig.1 and Fig.2) for clarity.
2. We added a new section 5.4 to discuss how convergence of the keep-learning VRM affect policy improvement.
3. We added Appendix F to empirically show the effect of hyperparameters of the VRMs on RL, which showed the representation learning part (training the VRMs) of our algorithm is robust to hyperparameters.
4. The scalability was explained in Appendix G.

Also some minor modifications on improving clarity and citing references.


(* Update after acceptance. *)
We thank the reviewers and the chairs for their time and efforts on reviewing our paper!  We here would like to clarify a mistake/typo in the figure descriptions we just found: In Fig.3 caption, we wrote "Shaded areas indicate 95% confidence interval.". However, it is actually “S.E.M.” instead of “95% confidence interval” (assuming Gaussian).  We are sorry for this mistake and have fixed it in the latest camera-ready version.

Also, the codes of our algorithm are now accessible at ( https://github.com/oist-cnru/Variational-Recurrent-Models ).

---

### Decision · Program_Chairs · 2019-12-19

**Decision:**

Accept (Poster)

**Comment:**

The authors propose to decompose control in a POMDP into learning a model of the environment (via a VRNN) and learning a feed-forward policy that has access to both the environment and environment model. They argue that learning the recurrent environment model is easier than learning a recurrent policy. They demonstrate improved performance over existing state-of-the-art approaches on several PO tasks.

Reviewers found the motivation for the proposed approach convincing and the experimental results proved the effectiveness of the method. The authors response resolved reviewers concerns, so as a result, I recommend acceptance.